# DIVA: A Dirichlet Process Mixtures Based Incremental Deep Clustering Algorithm via Variational Auto-Encoder

## Abstract

Generative model-based deep clustering frameworks excel in classifying complex data, but their effectiveness is limited when dealing with dynamically changing features due to their reliance on prior knowledge of the number of clusters. In this paper, we propose a nonparametric deep clustering framework that employs an infinite mixture of Gaussians as a prior. Our framework utilizes a memoized online variational inference method that enables the "birth" and "merge" moves of clusters, allowing our framework to cluster data in a "dynamic-adaptive" manner, without requiring prior knowledge of the number of features. We name the framework as *DIVA*, a *D*irichlet Process Mixtures based *I*ncremental deep clustering framework via *V*ariational *A*uto-Encoder. Our framework, which outperforms state-of-the-art baselines, exhibits superior performance in classifying complex data with dynamically changing features, particularly in the case of incremental features.

## 1 Introduction

Clustering is a key task in unsupervised learning that aims to group data points based on similarity or dissimilarity metrics. Recently, deep clustering algorithms that combine deep neural networks with clustering methods have shown great promise in various applications, such as image segmentation Zhou et al. (2022); Ren et al. (2022), document clustering Aggarwal & Zhai (2012); Yin & Wang (2014), and anomaly detection Kim et al. (2020). Generative model-based deep clustering algorithms have emerged as a promising research direction, with the variational auto-encoder (VAE) Kingma & Welling (2022) being a popular choice due to its ability to learn data representations and generation Ren et al. (2022). VAE-based clustering methods typically involve two stages: training a VAE to learn the underlying data distribution and then using the learned latent variables for clustering. The advantage of VAE is its ability to handle non-linear and complex distributions Zhou et al. (2022).

A natural idea in this field is to combine the Gaussian mixture model (GMM) Reynolds et al. (2009); Dilokthanakul et al. (2016); Zhou et al. (2022), which is a highly representative clustering module, with the VAE framework . This framework employs GMM as prior to provide with a richer information capacity, while the VAE's powerful representation learning and reconstruction capabilities can overcome the negative impact of the GMM shallow structure on the weak representation of non-linearity Zhou et al. (2022). However, such frameworks still have limitations, including the need to specify the number of clusters beforehand, which can be challenging when the number is unknown or varies across datasets. In Bayesian nonparametric field, previous work tried to replace the parameters of the isotropic Gaussian prior of standard VAE with the stick-breaking proportions of a Dirichlet process Nalisnick & Smyth (2016), where each latent dimension is a line segment of the stick and represents a cluster. However, these line segments only convey cluster membership information, lacking details about individual cluster shapes and densities. Additionally, embedding a Dirichlet process in the network architecture restricts variational inference to stochastic gradient variational Bayes, potentially introducing noise and leading to local optima.

To address this issue, we propose using the Dirichlet process mixture model (DPMM) from nonparametric Bayes as a clustering module for our framework. DPMM's random probability measure sampled from the base distribution through the stick-breaking process maintains both discrete and continuous characteristics, and the number of components can theoretically reach infinity, overcom-

ing the problem of pre-specifying the number of components. Additionally, we use the "birth" and "merge" behavior provided by DPMM for dynamic feature adaptation, allowing our framework to dynamically adjust the number of components according to the observed data to improve clustering performance. Our proposed framework demonstrates superior clustering performance and disentangled representation learning ability in various datasets, specially in facing incremental features, where the number of features in datasets gradually increases during training. This study presents novel insights on incorporating DPMM as a prior for the VAE and utilizing the "birth" and "merge" behavior to dynamically adjust the number of clusters in generative deep clustering framework.

The contributions of our paper are summarized as follows: First, we eliminate pre-defining the cluster number in prior space by introducing nonparametric clustering module DPMM into our VAE-based framework, allowing for clustering data with infinite features. Second, we introduce a memoized online variational Bayes inference method into the framework, which enables dynamic changes in the number, density, and shape of clusters in the prior space according to the observations. This allows for "birth" and "merge" of clusters. Third, we verify the dynamic-adaptation ability of our proposed DIVA, demonstrating its effectiveness against state-of-the-art baselines in facing incremental data features. We show that DIVA can dynamically adjust the clusters in the feature-increasing datasets (including real world images, texts or senors datasets) to maintain a high level of unsupervised clustering accuracy. The dynamic adaptation capability of DIVA demonstrated in our study has the potential to inspire new approaches to tackling the challenge of catastrophic forgetting Robins (1995), and could be extended to domains such as continuous learning De Lange et al. (2021), robotic motion planning Chen et al. (2023) and meta-reinforcement learning Bing et al. (2018).

## 2    RELATED WORK

Clustering constitutes a fundamental task in machine learning, and conventional models like $k$-means Jin & Han (2010) or GMM Reynolds et al. (2009) have been widely used. Moreover, Bayesian non-parametric models offer a more adaptable framework for handling an unknown number of features, with representative models such as the Dirichlet process mixture model Li et al. (2019), the Chinese Restaurant Process (CRP) Griffiths et al. (2003), and the Pitman-Yor Process (PYP) Qiang et al. (2018). Nevertheless, these clustering models exhibit limitations in dealing with complex high-dimensional data due to their shallow structure. To address this challenge, deep neural network-based clustering algorithms have emerged. In the Bayesian parametric domain, DEC Xie et al. (2016) and Deepcluster Tian et al. (2017) utilize $k$-means to estimate the similarity between feature embeddings and cluster centroids. Another approach involves combining generative models with GMMs in the prior space to cluster the latent representation, exemplified by methods VaDE Jiang et al. (2017) and GMVAE Dilokthanakul et al. (2016). Within the non-parametric domain, SB-VAE Nalisnick & Smyth (2016) adopts stick-breaking process as its prior space, enabling implicit clustering with infinite possibilities. VSB-DVM Yang et al. (2019) introduces density estimation structures to explicitly cluster data using stick-breaking as a latent regularizer. $\alpha JSD$ Lim (2021) leverages Jensen-Shannon divergence for both infinite model selection and deep clustering. DDPM Li et al. (2022a) and DeepDPM Ronen et al. (2022) utilize neural networks as feature extractors and sampling based methods to optimize parameters. VAE-nCRP Goyal et al. (2017) employs the nested Chinese Restaurant Process in its prior space, allowing for hierarchical video segmentation. Additionally, recent works have also integrated Hawkes processes with DPMM for clustering event or document sequences Xu & Zha (2017); Li et al. (2022b); Du et al. (2015); Lin (2013).

## 3    PRELIMINARIES

In this section, we introduce the concept of Bayesian nonparametric models and then the variational inference methods, which provide the theoretical foundation of our framework.

### 3.1    DIRICHLET PROCESS AND STICK-BREAKING METHOD

The Dirichlet process (DP) is a distribution over probability measures Teh (2010), where the marginal distribution is Dirichlet-distributed, resulting in random distributions. Given a base distribution $H$ and a positive concentration parameter $\alpha$, a random probability measure $G$ is DP-distributed, denoted as $G \sim \mathrm{DP}(\alpha, H)$ Hughes & Sudderth (2013); Teh (2010).

A DP can be defined constructively using the Stick-Breaking (SB) process Li et al. (2019) via Beta distribution $\mathcal{B}$, where for $k \geq 1$: $\beta_k \sim \mathcal{B}(1, \alpha)$, $\pi_k = \beta_k \prod_{i=1}^{k-1}(1 - \beta_i)$. In this process, a unit-length stick is imaginatively broken into an infinite number of segments $\pi_k$, with $\alpha$ being a positive scalar. We first sample $\beta_1 \sim \mathcal{B}(1, \alpha)$ from a Beta distribution and break the stick with length $\pi_1 = \beta_1$. We then sample $\beta_2$, and the length of the second segment will be $\pi_2 = \beta_2(1 - \beta_1)$. Continuing this process, we have $\sum_{k=1}^{\infty} \pi_k = 1$, and the resulting $\pi$ follows a Griffiths-Engen-McCloskey (GEM) distribution $\pi \sim \text{GEM}(\alpha)$. Figure 1a shows an intuitive image about the SB process.

Since a random distribution $G$ drawn from a DP maintains discrete property, it can be expressed as a weighted sum of point masses, namely $G = \sum_{k=1}^{\infty} \pi_k \delta_{\theta^*_k}$ Orbanz & Teh (2010), where $\delta_{\theta^*_k}$ is the point mass located at $\theta^*_k$: it equals 1 at $\theta^*_k$ and equals 0 elsewhere. By sampling the weights $\pi_k$ according to SB-process and sampling $\theta_k$ from a base distribution $\theta_k \sim H$, we can say that $G \sim \text{DP}(\alpha, H)$, indicating that $G$ is a Dirichlet Process with the base distribution $H$ and concentration parameter $\alpha$. Figure 1b shows a draw from a DP with $\alpha = 5$.

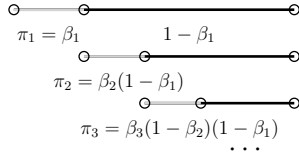

(a) Stick-breaking process.

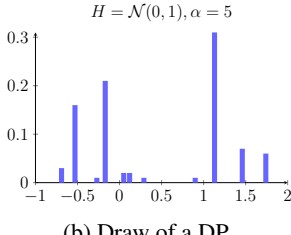

(b) Draw of a DP.

Figure 1: (a) Stick-breaking process. (b) Histogram of DP($\alpha = 5, H = \mathcal{N}(0, 1)$).

## 3.2 DIRICHLET PROCESS MIXTURE MODEL

DP mixture is a representative generative Bayesian nonparametric model that uses an infinite mixture of clusters to model a set of observations $\boldsymbol{x} = x_{1:N}$, where the number of cluster components is not predefined but rather determined by the observations.

The model assumes that each data point $x_i$ is sampled from a distribution $F(\theta_i)$ parameterized by a latent variable $\theta_i$ drawn independently from a probability measure $G$. The DPMM assumes a DP prior $G|\alpha, H \sim \text{DP}(\alpha, H)$, which introduces discreteness and clustering property where $\theta_i$ takes on repeated values. Then all $x_i$'s drawn with the same value of $\theta_i$ can be seen as one cluster, resulting in the clustering of observations. The number of unique values of $\theta_i$ determines the active number of cluster components, which can be dynamically inferred during inference from the observed data. Let $v_i$ be a cluster assignment variable that takes on value $k$ with probability $\pi_k$ drawn from a categorical distribution

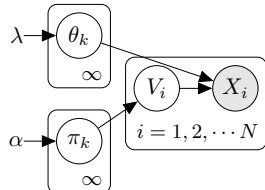

Figure 2: DPMM generative graphic model.

(Cat). The DPMM can be expressed via the stick-breaking process, where the mixing proportions $\pi$ are sampled from a GEM distribution and the prior $H$ over the cluster parameters is the base distribution of an underlying DP measure $G$. Specifically, we have:

$$\theta^*_k|H \sim H, \quad \pi|\alpha \sim \text{GEM}(\alpha), \quad v_i|\pi \sim \text{Cat}(\pi), \quad x_i|v_i \sim F(\theta^*_{v_i}), \tag{1}$$

where $\theta^*_k$ are the cluster parameters, $\pi$ is the mixing proportion, $F(\theta^*_{v_i})$ is the distribution over observation in cluster $k$, and $H$ is the prior over cluster parameters. Typically, $F$ is a Gaussian distribution. To provide an intuitive understanding, we draw a graphic model of DPMM in Figure 2.

## 3.3 VARIATIONAL INFERENCE FOR DPMM

In this section, we introduce variational inference as a method for approximating the posterior density for models based on observed data, with a focus on Bayesian nonparametric models. The basic idea of variational inference is to convert the inference problem into an optimization problem. Refer from equation 1, we write the joint probability for DPMM as:

$$p(\boldsymbol{x}, \boldsymbol{v}, \boldsymbol{\theta}, \boldsymbol{\beta}) = \prod_{n=1}^{N} F(x_n|\theta_{v_n})\text{Cat}(v_n|\boldsymbol{\pi}(\boldsymbol{\beta})) \prod_{k=1}^{\infty} \mathcal{B}(\beta_k|1, \alpha)H(\theta_k|\lambda). \tag{2}$$

Since the true posterior $p(\boldsymbol{v}, \boldsymbol{\theta}, \boldsymbol{\beta}|\boldsymbol{x})$ is intractable, we aim to find the best variational distribution $q(\boldsymbol{v}, \boldsymbol{\theta}, \boldsymbol{\beta})$ that minimizes the KL divergence with the exact conditional:

$$q^*(\boldsymbol{v}, \boldsymbol{\theta}, \boldsymbol{\beta}) = \arg\min_q \mathbb{KL}(q(\boldsymbol{v}, \boldsymbol{\theta}, \boldsymbol{\beta})||p(\boldsymbol{v}, \boldsymbol{\theta}, \boldsymbol{\beta}|\boldsymbol{x})),$$

$$\mathbb{KL}(q(\boldsymbol{v}, \boldsymbol{\theta}, \boldsymbol{\beta})||p(\boldsymbol{v}, \boldsymbol{\theta}, \boldsymbol{\beta}|\boldsymbol{x})) = \mathbb{E}[\log q(\boldsymbol{v}, \boldsymbol{\theta}, \boldsymbol{\beta})] - \mathbb{E}[\log p(\boldsymbol{x}, \boldsymbol{v}, \boldsymbol{\theta}, \boldsymbol{\beta})] + \log p(\boldsymbol{x}). \tag{3}$$

Notice that $\log p(\boldsymbol{x})$ does not depend on $q$, so instead of minimizing the KL divergence directly, we maximize the evidence lower bound (ELBO), which consists of the expected log-likelihood of the data $\mathbb{E}[\log p(\boldsymbol{x}|\boldsymbol{v}, \boldsymbol{\theta}, \boldsymbol{\beta})]$ and the KL divergence between two priors $\mathbb{KL}(q(\boldsymbol{v}, \boldsymbol{\theta}, \boldsymbol{\beta})||p(\boldsymbol{v}, \boldsymbol{\theta}, \boldsymbol{\beta}))$, according to the equation 3, the ELBO can be rewritten as:

$$\text{ELBO}(q) = \mathbb{E}[\log p(\boldsymbol{x}, \boldsymbol{v}, \boldsymbol{\theta}, \boldsymbol{\beta})] - \mathbb{E}[\log q(\boldsymbol{v}, \boldsymbol{\theta}, \boldsymbol{\beta})] = \mathbb{E}[\log p(\boldsymbol{x}|\boldsymbol{v}, \boldsymbol{\theta}, \boldsymbol{\beta})] - \mathbb{KL}(q(\boldsymbol{v}, \boldsymbol{\theta}, \boldsymbol{\beta})||p(\boldsymbol{v}, \boldsymbol{\theta}, \boldsymbol{\beta})). \tag{4}$$

Therefore, the optimization of the ELBO is interpreted as finding a solution that explains the observed data with minimal deviation from the prior.

For the DPMM model, based on the idea of variational inference, we define the variational distribution $q$ following the mean-field assumption, where each latent variable has its variational factor and is mutually independent of each other, namely $q(\boldsymbol{v}, \boldsymbol{\theta}, \boldsymbol{\beta}) = \prod_{n=1}^{N} q_{v_n} \prod_{k=1}^{K} q_{\beta_k} q_{\theta_k}$, with $q_{v_n} = \text{Cat}(v_n|\hat{r}_{n_1:n_K})$, $q_{\beta_k} = \mathcal{B}(\beta_k|\hat{\alpha}_{k_1}, \hat{\alpha}_{k_0})$, $q_{\theta_k} = H(\theta_k|\hat{\lambda}_k)$, where $q_{v_n}$ is categorical factor, $q_{\beta_k}$ is factor for stick-breaking proportion, and $q_{\theta_k}$ is base distribution factor. Letters with hats are corresponding variational parameters. In the context of variational inference, the true posterior distribution is infinite, and only an approximate distribution can be obtained. However, as the number of components $K$ in categorical factor increases, the optimized ELBO objective can result in a variational distribution that closely approximates the infinite posterior. Thus to enable computation, we limit the categorical factor to only finite $K$ components, in which $K$ is large enough to cover all potential features. We also consider a special case where the distributions $H$ and $F$ come from the exponential family. Hughes & Sudderth (2013) showed that in this case, the ELBO is expressed in terms of the expected mass $\hat{N}_k$ and the expected sufficient statistic $s_k(x)$ of each component $k$:

$$\text{ELBO}(q) = \sum_{k=1}^{K} \left[ \mathbb{E}_q[\theta_k]^T s_k(x) - \hat{N}_k[a(\theta_k)] + \hat{N}_k[\log \pi_k(\beta)] - \sum_{n=1}^{N} \hat{r}_{nk} \log \hat{r}_{nk} \right.$$
$$\left. + \mathbb{E}_q[\log \frac{\mathcal{B}(\beta_k|1, \alpha)}{q(\beta_k|\hat{\alpha}_{k_1}, \hat{\alpha}_{k_0})}] + \mathbb{E}_q[\log \frac{H(\theta_k|\lambda)}{q(\theta_k|\hat{\lambda}_k)}] \right]. \tag{5}$$

Then each variational factor can be updated in an iterative manner. In first stage, we update the *local* variational parameters $\hat{r}_{nk}$ in $q_{v_n}$ for each cluster assignment. In second stage, we update the *global* parameters $\hat{\alpha}_{k_1}, \hat{\alpha}_{k_0}, \hat{\lambda}_k$ in $q_{\beta_k}$ and $q_{\theta_k}$. The detail derivation can be found in appendix Sec.A.1.

The computation of the summary statistics $\hat{N}_k$ and $s_k(x)$ requires the full dataset. For inference on a large dataset, we use a batch-based approach called memoized online variational Bayes (memoVB) Hughes & Sudderth (2013), which breaks the summary statistics of full data into a sum of the summary statistics of each batch. The DPMM has nonparametric nature, which means it can adjust to varying numbers of clusters. This property enables the development of heuristics for dynamically adding or splitting clusters, preventing local optima in batch-based variational inference.

MemoVB is a heuristic approach that implements birth and merge moves for dynamic cluster adjustment. To create new clusters, we collect poorly described subsamples $x'$ by a single cluster when passing through each batch and fit a separate DPMM model with $K'$ initial clusters. Assuming that the active number of clusters before the birth move is $K$, we can either accept or reject the new cluster proposals by comparing the result of assigning $x'$ to $K + K'$ with that of assigning $x'$ to $K$. To complement the birth move, a merge move potentially combines a pair of clusters into one. Two clusters are merged if the merge improves the ELBO objective, leaving $K - 1$ clusters. For a comprehensive explanation of split and merge moves, please refer to the appendix in Sec. A.2.

## 4 METHODOLOGY

In this section, we present DIVA, a novel deep clustering approach that integrates a Bayesian nonparametric model with a variational autoencoder. Given an unlabeled set of data with $n$ points $\{x_i\}_{i=1}^{n} \in X$, the designed deep clustering method that simultaneously learns: (1) The number $K$ of Gaussian-distributed clusters and their associated means $\mu_k$ and covariance matrices $\Sigma_k$.

(2) The cluster assignment $v_i$ of each $x_i$, where $v_i \in \{1, \ldots, K\}$. (3) The latent representation $z_i$ of each $x_i$ in a $D$-dimensional Gaussian latent space $Z$ and the mapping $f_\theta : X \to Z$ that projects the data points $\{x_i\}_{i=1}^n \in X$ onto $Z$. (4) The $x_i^*$ via decoder network.

To accomplish this, DIVA combines a standard VAE with a DPMM+memoVB, where the cluster assignments are jointly determined by the learned representation and the cluster distributions. Our algorithm uses an alternating optimization scheme: First, we update the DPMM module using the latent variables $z_i$, which are sampled from the encoder during the last VAE update loop. Then the DPMM module is fixed, and the VAE parameters are updated, using the assigned clusters to each $z_i$ to minimize the KL divergence. An overview of DIVA is shown in Figure 3.

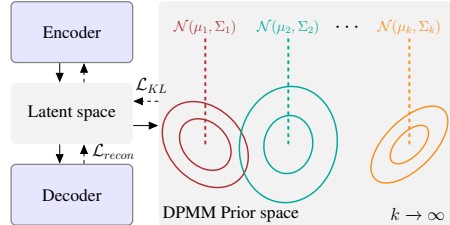

Figure 3: Overview of the DIVA. The DPMM and the VAE are optimized alternately. When updating the DPMM, we use the current latent sample $z$ obtained from the VAE. When updating the VAE, we assign the outputs of the encoder to the clusters of DPMM and minimize the $\mathcal{L}_{KL}$ with respect to the assigned cluster.

### 4.1 UPDATING THE DPMM

When updating the parameters of the DPMM, we fit a DPMM on the latent samples $z_i$'s obtained from the VAE. We define the generative process of assigning data points to clusters according to the SB process. A generative process is assumed as follows: (1) The mean $\mu_k$ and diagonal covariance matrix $\Sigma_k$ of each cluster $k$ are drawn from a Normal-Wishart distribution, which is the conjugate prior for a diagonal Gaussian with unknown means and unknown covariance matrix, namely $\Sigma_k \sim \mathcal{W}(\mathbf{W}, \nu)$, $\mu_k \sim \mathcal{N}(\mu_0, (\lambda \Sigma_k)^{-1})$, assuming the latent space is D-dimensional. (2) The probabilities of each cluster are drawn from a GEM distribution with concentration parameter $\alpha$, $\pi \sim \text{GEM}(\alpha)$, as described in Sec. 3.1. (3) The cluster assignment $v_n$ is drawn from a discrete distribution $\text{Cat}(\pi)$ based on the cluster probabilities $\pi$ previously obtained. (4) In our DPMM module, when the latent variable $z_i$ is assigned to a cluster $v_n = k$, we assume that it is sampled from a multivariate Gaussian with mean $\mu_k$ and variance $\Sigma_k$, which means $z_i | v_n = k \sim \mathcal{N}(\mu_k, \Sigma_k)$. (5) The original data are assumed to be generated by the decoder, namely $f_\theta(z_i) = x_i^*$, where $\theta$ are the parameters of the decoder. Given the overall generative process, the DPMM has a joint probability:

$$p(\boldsymbol{z}, \boldsymbol{v}, \boldsymbol{\beta}, \boldsymbol{\mu}, \Sigma) = \prod_{i=1}^{N} \mathcal{N}(z_i | \mu_{v_n}, \Sigma_{v_n}) \text{Cat}(v_n | \boldsymbol{\pi}(\boldsymbol{\beta})) \prod_{k=1}^{\infty} \mathcal{B}(\beta_k | 1, \alpha) \prod_{k=1}^{\infty} \mathcal{N}(\mu_k | \mu_0, (\lambda \Sigma_k)^{-1}) \mathcal{W}(\Sigma_k | \mathbf{W}, \nu). \quad (6)$$

We then use variational inference to find the posterior estimates for the parameters. We construct the variational distribution $q$ with the following factorization:

$$q(\boldsymbol{v}, \boldsymbol{\beta}, \boldsymbol{\mu}, \Sigma) = \prod_{n=1}^{N} \text{Cat}(v_n | \hat{r}_{n_1}, \cdots, \hat{r}_{n_k}) \prod_{k=1}^{K} \mathcal{B}(\beta_k | \hat{\alpha}_{k_1}, \hat{\alpha}_{k_0}) \prod_{k=1}^{K} \mathcal{N}(\mu_k | \hat{\mu}_0, (\hat{\lambda} \Sigma_k)^{-1}) \mathcal{W}(\Sigma_k | \hat{\mathcal{W}}, \hat{\nu}). \quad (7)$$

We propose the iterative optimization procedure described in Sec. 3.3 that updates the Normal-Wishart distribution parameters $\hat{\mu}_0$, $\hat{\lambda}$, $\hat{\mathcal{W}}$, and $\hat{\nu}$ in each step. We also utilize the memoized online variational Bayes method to update the parameters of the stick-breaking process $\hat{r}_{n_1:n_k}$, $\hat{\alpha}_{k_1}$, and $\hat{\alpha}_{k_0}$ to estimate the cluster probabilities and assignments. In addition, we dynamically adjust the total number of clusters $K$ using birth-and-merge moves. A complete update for our DPMM module thus breaks down to updating three sets of parameters.

Each update of the DPMM module is performed after a training epoch of the VAE. Since we alternate between updating the DPMM module and the VAE, the DPMM module is not required to converge in each update and avoids the need to fit a new DPMM model from scratch every time. In each update, we initialize the DPMM with the parameters learned from the previous updates and apply this module to new latent samples produced by the updated VAE, enabling us to update the same DPMM while incorporating the latest changes in the latent space mappings.

### 4.2 UPDATING THE VAE

When training the VAE, we jointly minimize the reconstruction loss $\mathcal{L}_{recon}$ and the KL divergence loss $\mathcal{L}_{KL}$. The reconstruction loss $\mathcal{L}_{recon}$ is the mean squared error between the observed data $x$ and the decoder's reconstructions $x^*$, which is kept from standard VAE. The $\mathcal{L}_{KL}$ is the KL-divergence between two isotropic Gaussian Kingma & Welling (2022). To compute $\mathcal{L}_{KL}$, we obtain

the cluster assignment $v_i = k$ of each latent sample $z_i$ from the current DPMM model. Using the DPMM, the mean and covariance matrix of assigned cluster $k$ is $\mu_k$ and $\Sigma_k$. According to VAE, $z_{ik} = \mu_k(\boldsymbol{x}; \boldsymbol{\phi}) + \Sigma_k(\boldsymbol{x}; \boldsymbol{\phi})\eta$ for $k \in \{1, 2, \ldots, K\}$, where $\phi$ are encoder parameters, $\mu(x_i; \boldsymbol{\phi})$ and $\Sigma(x_i; \boldsymbol{\phi})$ are encoder outputs and $\eta \sim \mathcal{N}(0, 1)$. The KL divergence between two multivariate Gaussian distributions is calculated as follows:

$$\mathcal{L}_{\mathrm{KL}_{ik}} = \frac{1}{2}\left[ \log \frac{|\Sigma_k|}{|\Sigma(x_i; \boldsymbol{\phi})|} - D + \mathrm{tr}\{\Sigma_k^{-1}\Sigma(x_i; \boldsymbol{\phi})\} + (\mu_k - \mu(x_i; \boldsymbol{\phi}))^T \Sigma_k^{-1}(\mu_k - \mu(x_i; \boldsymbol{\phi})) \right]. \quad (8)$$

However, this hard assignment method may assign a sample to a wrong cluster, introducing errors into the training by calculating the KL divergence. To overcome this issue, we introduce *soft assignment*, in which we compute the probability $p_{ik}$ of assigning the latent sample $z_i$ to cluster $k$ using the DPMM for all possible $k \in \{1, 2, \ldots, K\}$. Then the KL divergence of i-th sample is defined as a weighted sum with respect to each cluster component:

$$\mathcal{L}_{\mathrm{KL}_i} = \sum_{k=1}^{K} p_{ik}\mathcal{L}_{\mathrm{KL}_{ik}}. \quad (9)$$

Although one can use a more complex weighting strategy, we found this simple weighting by probabilities to be sufficient empirically. We use Algorithm 1 to summarize the DIVA algorithm.

---

**Algorithm 1** DIVA

---

**Require:** Dataset $\mathcal{D}$, batch size $B$, DPMM train steps $T$, parameters $\phi$ of the encoder and $\theta$ of the decoder.
1: Initialize $\phi, \theta$ of VAE, the DPMM with $K = 1$, including $\mu_0, \lambda, \mathcal{W}, \nu$ and $\alpha$.
2: **repeat**
3:     Sample mini-batch $\mathcal{M} = \{x_{0:B}\} \sim \mathcal{D}$.
4:     With the VAE, obtain latent variables $z_{0:B}$ and save to a buffer $\mathcal{Z} = \mathcal{Z} \bigcup \{z_{0:B}\}$.
5:     With the current DPMM, obtain the cluster assignments $v_{0:B}$ and the assignment probabilities of each latent variable.
6:     Compute $\mathcal{L}_{KL}$ using equation 8, equation 9, $\mathcal{L}_{recon}$ and update $\phi, \theta$.
7:     **if** time to update the DPMM **then**
8:         **for** step $i = 1, 2, ..., T$ **do**
9:             Update DPMM variables $\hat{\mu}_0, \hat{\lambda}, \hat{\mathcal{W}}, \hat{\nu}, \hat{r}_{n_1:n_k}, \hat{\alpha}_{k_1}, \hat{\alpha}_{k_0}$ using $\mathcal{Z}$ and current DPMM.
10:             birth moves to try to add new clusters, and merge moves to try to combine existing clusters.
11:     Reset buffer $\mathcal{Z} = \emptyset$.
12: **until** convergence.

---

## 5 EXPERIMENTS

### 5.1 IMPLEMENTATION DETAILS

To assess the scalability of our proposed framework, we conduct evaluations on eight widely adopted datasets, including MNIST LeCun et al. (2012), Fashion-MNIST Xiao et al. (2017), Reuters10k Jiang et al. (2017), HHAR Li et al. (2022a), as well as four more challenging real-world image datasets: STL-10 Coates et al. (2011), ImageNet-50 Deng et al. (2009), CIFAR-10 Lim (2021), and SVHN Netzer et al. (2011). Directly learning from STL-10 and ImageNet can be challenging, therefore, we apply feature pre-extraction from previous studies Jiang et al. (2017); Ronen et al. (2022). Specifically, we utilize pretrained ResNet-50 He et al. (2015) and MOCO Chen et al. (2020) models to reduce the learning burden when handling raw images from STL-10 and ImageNet. On CIFAR-10 and SVHN, we evaluate the generative capabilities of our model using the raw images.

Our comparative analysis involved eight state-of-the-art (SOTA) baselines, including three parametric Bayesian methods: GMM Reynolds et al. (2009), Deep Embedded Clustering (DEC) Xie et al. (2016), and Gaussian Mixture Variational Autoencoder (GMVAE) Dilokthanakul et al. (2016). Additionally, we consider five non-parametric approaches: memoVB Hughes & Sudderth (2013), VSB-DVM Yang et al. (2019), DDPM Li et al. (2022a), DeepDPM Ronen et al. (2022), and SB-VAE Nalisnick & Smyth (2016). In our framework, we employ a 2-layer CNN network as the backbone for MNIST and Fashion MNIST datasets. For the remaining datasets, we use a MLP with two hidden layers, having dimensions d-500-500-2000-k, where d and k represent the input and output dimensions,

respectively. In the case of the SOTA baselines, we directly utilize available open-source code with minor adjustments to maintain consistent comparisons, such as modifying random seeds and training epochs. However, we exclude $\alpha JSD$ and VAE-nCRP from our study due to the unavailability of open-source code and different datasets they used. For further implementation details, we refer the reader to the appendix in Section A.3.

To assess the unsupervised clustering performance, we utilize clustering accuracy (ACC), a key metric adopted from prior studies Xie et al. (2016); Dilokthanakul et al. (2016). We report ACC performance in main page, other metrics results including Normalized Mutual Information (NMI) and Adjusted Rand Index (ARI), please refer to appendix Sec. A.4.2.

## 5.2 EXPERIMENTAL RESULTS ON COMPLETE STATIC DATASETS

The ACC results of complete static datasets are presented in Table 1. We conduct trials on all datasets, running them up to 500 epochs or until the frameworks converge. Each trial is repeated five times, and we report their average performance on test datasets, combined with the standard deviation for final evaluation. We highlight the best two performance achieved on corresponding datasets. Additionally, we consider an imbalanced Reuters10k dataset, with a ratio between the maximal and minimal class groups of $43\% : 8\%$. It is worth noting that our DIVA framework outperforms both parametric and non-parametric SOTA baselines on all benchmark datasets. Even on the most complex ImageNet-50 dataset with 50 classes, our framework can achieve an ACC of up to $0.69$. Notably, recent studies Dilokthanakul et al. (2016); Xie et al. (2016); Ronen et al. (2022) primarily concentrate on evaluating performance within static scenarios, wherein the number of features may be unknown but remain unchanged during training. The advantages of our framework are more apparent when handling dynamic changing features, as evidenced in the subsequent Sec. 5.3.

Table 1: ACC comparison on complete static datasets. mean $\pm$ (std.dev.) of 5 runs, with higher values indicating better performance. †: Trials executed using the authors' open-source code. $imb$: Imbalanced category proportion. ‡: Results sourced from Xie et al. (2016).

| Frameworks | MNIST | Fashion-MNIST | Reuters10k$^{imb}$ | HHAR | STL-10 | ImageNet-50 |
|---|---|---|---|---|---|---|
| GMM | $.60 \pm .01$ | $.49 \pm .02$ | $.73 \pm .06$ | $.43 \pm .00$ | $.58 \pm .03$ | $.60 \pm .01$ |
| DEC† | $.84 \pm .00^{\ddagger}$ | $.60 \pm .04$ | $.72 \pm .00^{\ddagger}$ | $\mathbf{.79} \pm \mathbf{.01}$ | $.80 \pm .01$ | $.63 \pm .01$ |
| GMVAE† | $.82 \pm .04$ | $.61 \pm .01$ | $.73 \pm .08$ | $.65 \pm .03$ | $.79 \pm .04$ | $.62 \pm .02$ |
| DPMM+memoVB | $.63 \pm .02$ | $.57 \pm .01$ | $.56 \pm .05$ | $.68 \pm .04$ | $.64 \pm .05$ | $.57 \pm .00$ |
| VSB-DVM† | $.86 \pm .01$ | $\mathbf{.64} \pm \mathbf{.03}$ | $.60 \pm .03$ | $.66 \pm .06$ | $.52 \pm .03$ | $.49 \pm .02$ |
| DDPM† | $.91 \pm .01$ | $.63 \pm .02$ | $.71 \pm .02$ | $.74 \pm .01$ | $.72 \pm .01$ | $.63 \pm .02$ |
| DeepDPM† | $\mathbf{.93} \pm \mathbf{.02}$ | $.63 \pm .01$ | $\mathbf{.83} \pm \mathbf{.01}$ | $.79 \pm .02$ | $\mathbf{.81} \pm \mathbf{.02}$ | $\mathbf{.66} \pm \mathbf{.01}$ |
| **DIVA (Ours)** | $\mathbf{.94} \pm \mathbf{.01}$ | $\mathbf{.72} \pm \mathbf{.01}$ | $\mathbf{.83} \pm \mathbf{.01}$ | $\mathbf{.83} \pm \mathbf{.01}$ | $\mathbf{.88} \pm \mathbf{.01}$ | $\mathbf{.69} \pm \mathbf{.02}$ |

Additionally, we present the t-SNE Cieslak et al. (2020) projection of the full MNIST in Figure 4 for both DIVA and some of the baselines, colored by the ground truth. As shown in the figures, our DIVA is capable of learning a distinct and cluster-friendly representation, while other baseline methods, such as SB-VAE, fails to learn a latent space with high discrimination. GMVAE with a proper number of clusters ($K = 10$) (Fig. 4e), can learn a better clustering representation, but some clustering groups are still "stick" together. However, when the number of defined clusters is smaller than the number of features, GMVAE fails to learn a distinct latent representation (Fig. 4f, 4g).

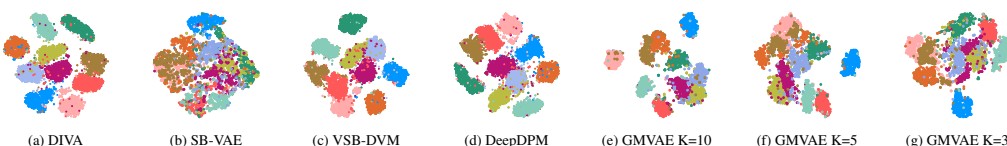

(a) DIVA    (b) SB-VAE    (c) VSB-DVM    (d) DeepDPM    (e) GMVAE K=10    (f) GMVAE K=5    (g) GMVAE K=3

Figure 4: t-SNE projection on full static MNIST, colored by the ground truth. It is clearly to see that DIVA can learn a clustered latent representation with high distinction. GMVAE with improper defined cluster number (Fig. 4f, 4g) can not learn a distinct clustering representation. Notably, the advantages of our framework are more apparent when handling dynamic changing features.

## 5.3 INCREMENTAL REPRESENTATION LEARNING AND CLUSTERING

**Outperforming parametric models.** To demonstrate the superior performance of our non-parametric framework DIVA compared to parametric baselines when dealing with an increasing number of features, Figure 5 displays the clustering accuracy achieved by DIVA and existing parametric models on three datasets with increased features: (a) MNIST, (b) Fashion-MNIST, and (c) STL-10. We train the models on datasets with a fixed number of ground truth classes and gradually increase the class number in the subsequent trials. The bar plot represents the mean values of clustering accuracy along with the corresponding standard deviation. Our framework consistently achieves high clustering accuracy, with an average accuracy of 0.94 and 0.72 on MNIST and Fashion-MNIST, respectively, and up to 0.88 on STL-10. Notably, our approach outperforms the parametric baselines GMVAE and GMM that require prior specification of the number of clusters. In the case of larger feature sets than the number of pre-defined clusters, the clustering accuracy of these baseline models significantly declines due to its inherent limitation about unchangeable cluster number in prior space.

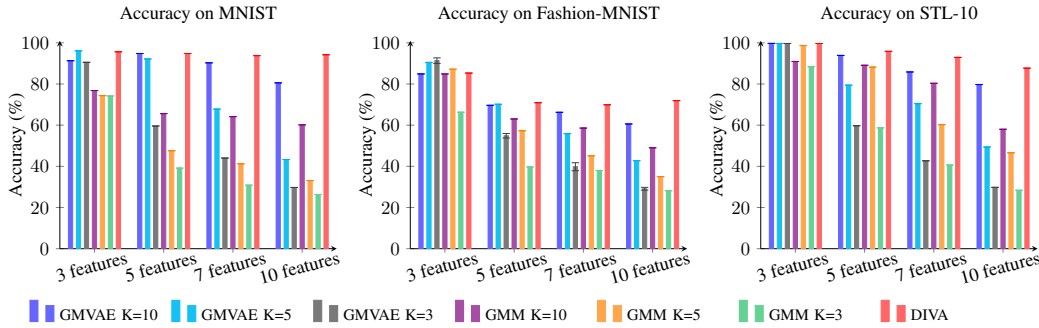

Figure 5: Clustering accuracy with incremental features for MNIST (left), Fashion-MNIST (middle) and STL-10 (right). mean $\pm$ (std.dev.) of 5 runs. We evaluate our framework and parametric baselines on test dataset with incremental number of features, e.g., for MNIST the x-axis with "3 features" means the dataset contains 3 types of digit to be classified, which are "0, 1, 2", respectively.

**Dynamic adaptation.** We simulate a complex real-world scenario where models learn from a data stream with a gradual increase in features. This requires the models to detect and adapt to new inputs in a few-shot or even zero-shot manner. We utilize the MNIST dataset for our evaluation. Initially, the models are trained with 3 digits, and at epochs 30, 60, and 90, we increment the number of digits to 5, 7, and 10, respectively. Notably, DDPM and DeepDPM require pretrained extractors on corresponding tasks, typically involving hundreds of training epochs before clustering. Consequently, they cannot adapt to changes even in a few-shot manner. Given DeepDPM's competitive performance in static clustering, we select and modify it for our comparisons. To alleviate the learning burden of such extractors and sampling-based methods, we pre-extract all data using pretrained extractors before training. We then employ these extracted encodings for clustering, whereas other frameworks directly use the raw images. Figure 6 depicts the test ACC of all selected models (solid lines) and the number of cluster changes in DIVA (dashed line). As new features are introduced to the dataset, DIVA dynamically creates new components to accommodate them, potentially leading to a temporary decrease in accuracy during early training stages when the VAE and DPMM have not yet converged. However, once training on the new subset converges, accuracy returns to the highest possible level. In contrast, GMVAE lacks this dynamic-adaptive capability. When the number of features surpasses the number of components, its test accuracy exhibits a stepwise decline. VSB-DVM achieves similar result as GMVAE K=10. Vanilla DPMM+memoVB cannot handle high-dimensional raw data and thus performs worse. Notably, in this scenario, DeepDPM cannot classify the inputs or generate new components to accommodate them, even with extracted encodings. This may be attributed to the limited sample efficiency of the clustering module. Additional trial results please refer to the appendix Sec. A.4.3 and video visualization. The right plot in Figure 6 shows DIVA's learned latent space projection, colored by clusters. Notably, each ground truth is learned by 2 or 3 sub-clusters, resulting in a total number of components greater than the number of ground truth. This arises from DPMM's ability to capture not only coarse labels but also the sub-features within each class, which is demonstrated through visualization results in the subsequent generative performance part.

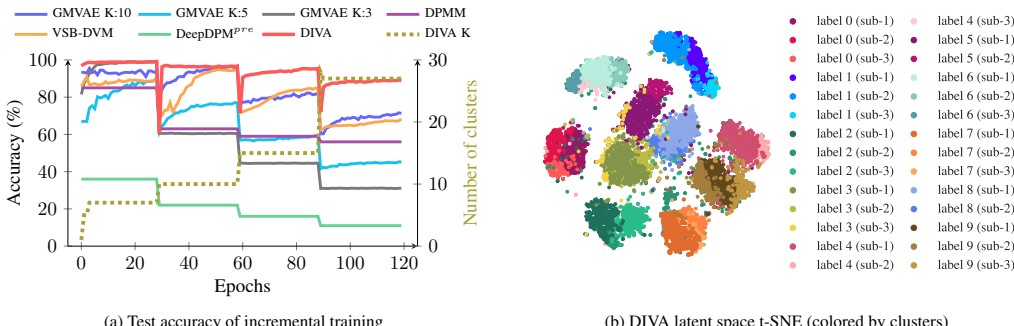

(a) Test accuracy of incremental training

(b) DIVA latent space t-SNE (colored by clusters)

Figure 6: (a) Zero-shot adaptation performance on MNIST. We introduce dynamic feature in training, where the model is initially trained on 3 digits, and the number of digits is increased to 5, 7, and 10 at epochs 30, 60, and 90. We record the test accuracy (solid lines) and the number of clusters for DIVA (dashed line). (b) t-SNE projection of DIVA on MNIST at last epoch, colored by clusters. Each coarse label is learned by 2 or 3 clusters, enabling the sub-features of individual digits to be captured. $pre$: pre-extract raw data to reduce training burden.

**Generative performance.** To gain more insight into what individual sub-cluster has learned, we conducted image reconstructions using the learned components in the DPMM. Figure 7 presents the generated images from sub-clusters on MNIST (a)-(c), (j)-(l); Fashion-MNIST (d)-(f), (m)-(o); CIFAR-10 (g)-(i), and SVHN (p)-(r). The illustrations in Figure 7 showcase that DPMM components cluster both coarse labels and sub-features within the ground truth in a hierarchical manner. For instance, in the case of MNIST, the three sub-clusters in DPMM capture the digit "0" in (a)-(c) while also learning different writing styles of "0". On CIFAR-10, the clusters capture not only the object but also the color styles of the images. Overall, our proposed DIVA can efficiently extract informative sub-features from coarse labels. Specifically, when dealing with unknown number of features, DIVA's disentangled representation learning capability is highly beneficial in uncovering deep information from data samples.

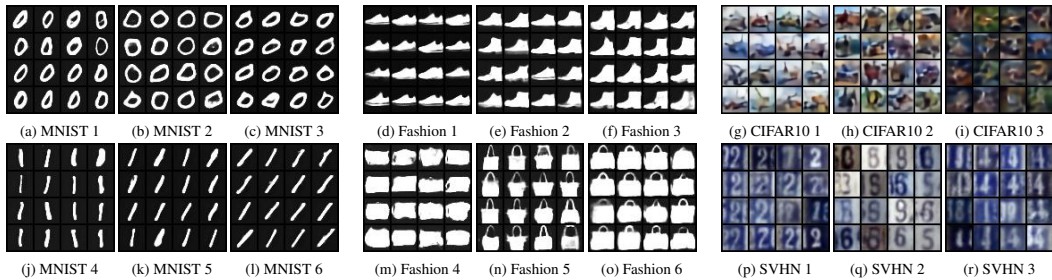

| (a) MNIST 1 | (b) MNIST 2 | (c) MNIST 3 | (d) Fashion 1 | (e) Fashion 2 | (f) Fashion 3 | (g) CIFAR10 1 | (h) CIFAR10 2 | (i) CIFAR10 3 |

| (j) MNIST 4 | (k) MNIST 5 | (l) MNIST 6 | (m) Fashion 4 | (n) Fashion 5 | (o) Fashion 6 | (p) SVHN 1 | (q) SVHN 2 | (r) SVHN 3 |

Figure 7: Reconstruction images from DPMM learned clusters on MNIST (a)-(c), (j)-(l); Fashion-MNIST (d)-(f), (m)-(o); CIFAR-10 (g)-(i) and SVHN (p)-(r). Each subfigure with 16 plots stems from one cluster. It is noting that our proposed framework DIVA can efficiently extract informative sub-features from coarse labels. More results refer to appendix Sec. A.4.4.

## 6 CONCLUSION

Our proposed framework, DIVA, utilizes the infinite clustering property of Bayesian non-parametric mixtures and combines it with the powerful latent representation learning ability of VAEs to overcome the challenge of clustering complex or dynamically changing data without prior knowledge of the feature number. The dynamic adaptation exhibited by DIVA on three datasets outperforms state-of-the-art baselines in handling data with incremental features. In addition, our framework excels in discovering finer-grained features and its adaptability to observed data suggests potential applications in domains like continuous learning. We encourage readers to explore further extensions and applications based on our framework.

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

## A    Supplementary Material

### A.1    Variational inference of Dirichlet process mixture model

We draw generative graphic models for both GMM and DPMM in supplementary Figure 1 to provide an intuitive understanding about the generative process of DP mixture.

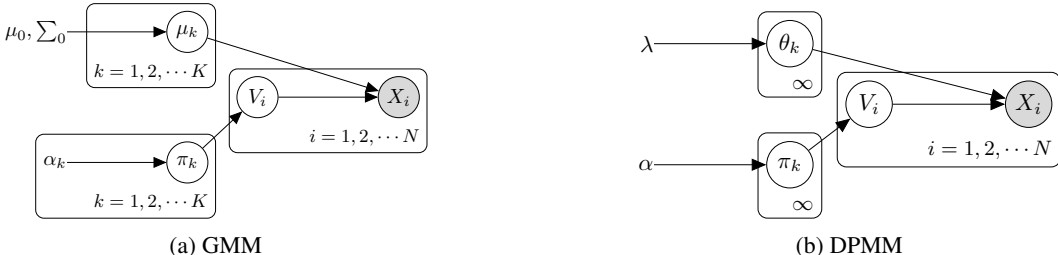

(a) GMM                                    (b) DPMM

Figure 1: Generative graphic model of (a) the Gaussian mixture model and (b) the Dirichlet process mixture model. Compared with GMM that has fixed numbers of clusters, DPMM can model an infinite number of clusters.

Given a set of observations $\boldsymbol{x} = \{x_1, ..., x_N\}$, recall that we can represent a Gaussian mixture model (GMM) as a generative model as follows:

$$
\begin{aligned}
\mu_i &\sim \mathcal{N}(\mu_0, \Sigma_0), \\
\pi_{1:K} &\sim Dirichlet(\alpha_{1:K}), \\
v_i &\sim Cat(\pi_{1:K}), \\
x_i | v_i &\sim \mathcal{N}(\mu_{v_i}, \Sigma),
\end{aligned}
\tag{1}
$$

where each $v_i$ represents a latent cluster for $x_i$ and $\pi$'s are the mixing proportions. The Dirichlet process mixture model (DPMM) is an infinite mixture model from Bayes non-parametrics where the number of cluster components is decided by the data instead of being pre-specified. A set of observations $\boldsymbol{x} = \{x_1, ..., x_N\}$ is modeled by a set of latent parameters $\boldsymbol{\theta} = \{\theta_1, ..., \theta_N\}$, where $\theta_i$ is draw from $G$. Here $x_i$ is assumed to be described by a component distribution $F(\theta_i)$ parameterized by $\theta_i$. Then we have:

$$
\begin{aligned}
G | \alpha, H &\sim \text{DP}(\alpha, H), \\
\theta_i | G &\sim G, \\
x_i | \theta_i &\sim F(\theta_i).
\end{aligned}
\tag{2}
$$

Meanwhile, the Dirichlet process can be alternatively represented using the stick-breaking process. Given $\alpha$ and $H$, we express $G | \alpha, H \sim \text{DP}(\alpha, H)$ as follows:

- Sample cluster parameters $\theta_k^* | H \sim H$ from the base distribution $H$.
- Sample the corresponding mixture proportions $\pi | \alpha \sim \text{GEM}(\alpha)$.
- Multiply the cluster parameters and mixture proportions to obtain a random discrete probability measure from the Dirichlet process, denoted as $G$.
- Use categorical distribution with mixture proportion $\pi$ to get the cluster assignment, and corresponding cluster parameters $\theta_{v_i}^*$.

The generative process is further explained in Section 3.2 and summarized below.

$$
\begin{aligned}
\theta_k^* | H &\sim H, \\
\pi | \alpha &\sim \text{GEM}(\alpha), \\
v_i | \pi &\sim \text{Cat}(\pi), \\
x_i | v_i &\sim F(\theta_{v_i}^*).
\end{aligned}
\tag{3}
$$

According to the generative process equation 3, the joint probability of DPMM $p(\boldsymbol{x}, \boldsymbol{v}, \boldsymbol{\theta}, \boldsymbol{\beta})$ is written as:

$$p(\boldsymbol{x}, \boldsymbol{v}, \boldsymbol{\theta}, \boldsymbol{\beta}) = \prod_{n=1}^{N} F(x_n|\theta_{v_n})\text{Cat}(v_n|\boldsymbol{\pi}(\boldsymbol{\beta})) \prod_{k=1}^{\infty} \mathcal{B}(\beta_k|1, \alpha)H(\theta_k|\lambda). \tag{4}$$

Where $\theta_{v_n}$ is component distribution parameter, $\alpha$ is stick-breaking proportion parameter and $\lambda$ is the parameters of base distribution. Since true posterior $p(\boldsymbol{v}, \boldsymbol{\theta}, \boldsymbol{\beta}|\boldsymbol{x})$ is intractable, thus we aim to find best distribution that minimize the KL divergence, namely:

$$q^*(\boldsymbol{v}, \boldsymbol{\theta}, \boldsymbol{\beta}) = \arg\min_{q} \mathbb{KL}(q(\boldsymbol{v}, \boldsymbol{\theta}, \boldsymbol{\beta})||p(\boldsymbol{v}, \boldsymbol{\theta}, \boldsymbol{\beta}|\boldsymbol{x})), \tag{5}$$

$$\begin{aligned}
\mathbb{KL}(q(\boldsymbol{v}, \boldsymbol{\theta}, \boldsymbol{\beta})||p(\boldsymbol{v}, \boldsymbol{\theta}, \boldsymbol{\beta}|\boldsymbol{x})) &= \mathbb{E}[\log q(\boldsymbol{v}, \boldsymbol{\theta}, \boldsymbol{\beta})] - \mathbb{E}[\log p(\boldsymbol{v}, \boldsymbol{\theta}, \boldsymbol{\beta}|\boldsymbol{x})] \\
&= \mathbb{E}[\log q(\boldsymbol{v}, \boldsymbol{\theta}, \boldsymbol{\beta})] - \mathbb{E}[\log p(\boldsymbol{x}, \boldsymbol{v}, \boldsymbol{\theta}, \boldsymbol{\beta})] \\
&\quad + \log p(\boldsymbol{x}).
\end{aligned} \tag{6}$$

$$\log p(\boldsymbol{x}) - \mathbb{KL}(q(\boldsymbol{v}, \boldsymbol{\theta}, \boldsymbol{\beta})||p(\boldsymbol{v}, \boldsymbol{\theta}, \boldsymbol{\beta}|\boldsymbol{x})) = \mathbb{E}[\log p(\boldsymbol{x}, \boldsymbol{v}, \boldsymbol{\theta}, \boldsymbol{\beta})] - \mathbb{E}[\log q(\boldsymbol{v}, \boldsymbol{\theta}, \boldsymbol{\beta})]. \tag{7}$$

Since $\log p(\boldsymbol{x})$ is not depend on $q$, according to the variational inference theory, this is equivalent to maximize the ELBO of $q$, which is

$$\begin{aligned}
\text{ELBO}(q) &= \mathbb{E}[\log p(\boldsymbol{x}, \boldsymbol{v}, \boldsymbol{\theta}, \boldsymbol{\beta})] - \mathbb{E}[\log q(\boldsymbol{v}, \boldsymbol{\theta}, \boldsymbol{\beta})] \\
&= \mathbb{E}[\log p(\boldsymbol{v}, \boldsymbol{\theta}, \boldsymbol{\beta})] + \mathbb{E}[\log p(\boldsymbol{x}|\boldsymbol{v}, \boldsymbol{\theta}, \boldsymbol{\beta})] - \mathbb{E}[\log q(\boldsymbol{v}, \boldsymbol{\theta}, \boldsymbol{\beta})] \\
&= \mathbb{E}[\log p(\boldsymbol{x}|\boldsymbol{v}, \boldsymbol{\theta}, \boldsymbol{\beta})] - \mathbb{KL}(q(\boldsymbol{v}, \boldsymbol{\theta}, \boldsymbol{\beta})||p(\boldsymbol{v}, \boldsymbol{\theta}, \boldsymbol{\beta})).
\end{aligned} \tag{8}$$

The initial component, $\mathbb{E}[\log p(\boldsymbol{x}|\boldsymbol{v}, \boldsymbol{\theta}, \boldsymbol{\beta})]$, represents the anticipated log-likelihood of the data. This term prompts the variational distribution to lean towards parameter values that effectively elucidate the observed data. The second constituent, $\mathbb{KL}(q(\boldsymbol{v}, \boldsymbol{\theta}, \boldsymbol{\beta})||p(\boldsymbol{v}, \boldsymbol{\theta}, \boldsymbol{\beta}))$, encapsulates the KL divergence between two priors: $p(\boldsymbol{v})$ and $q(\boldsymbol{v})$. This divergence compels the variational distribution to remain proximate to the prior. Hence, we can conceive of the optimization of the ELBO as an endeavor to discover a solution that adequately explains the observed data while minimizing divergence from the prior distribution. To enable computation of $q(\boldsymbol{v}, \boldsymbol{\theta}, \boldsymbol{\beta})$, here we assume that $q$ follows mean-field assumption, where each latent variable has its variational factor and is mutually independent of each other.

$$q(\boldsymbol{v}, \boldsymbol{\theta}, \boldsymbol{\beta}) = \prod_{n=1}^{N} q(v_n|\hat{r}_n) \prod_{k=1}^{K} q(\beta_k|\hat{\alpha}_{k_1}, \hat{\alpha}_{k_0})q(\theta_k|\hat{\lambda}_k), \tag{9}$$

$$\begin{aligned}
q_{v_n} &= \text{Cat}(v_n|\hat{r}_{n_1:n_K}), \\
q_{\beta_k} &= \mathcal{B}(\beta_k|\hat{\alpha}_{k_1}, \hat{\alpha}_{k_0}), \\
q_{\theta_k} &= H(\theta_k|\hat{\lambda}_k).
\end{aligned} \tag{10}$$

Where $q_{v_n}$ is categorical factor, $q_{\beta_k}$ stick-breaking proportion factor and $q_{\theta_k}$ is factor of base distribution $H$. In order to distinguish the parameters of the variational factors $q$ from those of the generative model $p$, we denote the former with hats. The categorical factor $q_{v_n}$ is constrained to a maximum of $K$ components to enable efficient computation. However, as the variational distribution is merely an approximation, the true posterior remains infinite. By increasing the value of $K$, the variational distribution can approach the infinite posterior through the optimization of the ELBO.

In addition, we consider a special case where both base distribution $H$ and each component distribution $F$ in equation 4 belong to the exponential family:

$$p_H(\theta_k|\lambda_0) = \mathbb{E}[\lambda_0^T t_0(\theta_k) - a_0(\lambda_0)], \tag{11}$$

$$p(x_n|\theta_k) = \mathbb{E}[\theta_k^T t(x_n) - a(\theta_k)]. \tag{12}$$

In this case, the ELBO can be expressed in terms of the expected mass $\hat{N}_k$ and the expected sufficient statistic $s_k(x)$ of each component $k$ Hughes & Sudderth (2013):

$$\begin{aligned}
\text{ELBO}(q) = \sum_{k=1}^{K} &\left[ \mathbb{E}_q[\theta_k]^T s_k(x) - \hat{N}_k[a(\theta_k)] + \hat{N}_k[\log \pi_k(\beta)] - \sum_{n=1}^{N} \hat{r}_{nk} \log \hat{r}_{nk} \right. \\
&\left. + \mathbb{E}_q[\log \frac{\mathcal{B}(\beta_k|1, \alpha)}{q(\beta_k|\hat{\alpha}_{k_1}, \hat{\alpha}_{k_0})}] + \mathbb{E}_q[\log \frac{H(\theta_k|\lambda)}{q(\theta_k|\hat{\lambda}_k)}] \right].
\end{aligned} \tag{13}$$

Each variational factor can be updated iteratively in a sequential manner. The first step involves updating the *local* variational parameters for the categorical factor $q_{v_n}$ associated with each cluster assignment:

$$\tilde{r}_{nk} = \exp(\mathbb{E}_q[\log \pi_k(\beta)] + \mathbb{E}_q[\log p(x_n|\theta_k)]), \tag{14}$$

$$\hat{r}_{nk} = \frac{\tilde{r}_{nk}}{\sum_{l=1}^{K} \hat{r}_{nl}}. \tag{15}$$

Subsequently, we proceed to update the *global* parameters in the stick-breaking proportions factor $q_{\beta_k}$ and the base distribution factor $q_{\theta_k}$.

$$\hat{N}_k = \sum_{n=1}^{N} \hat{r}_{nk}, \qquad \alpha_{k1} = \alpha_1 + \hat{N}_k, \qquad \alpha_{k0} = \alpha_0 + \sum_{l=k+1}^{N} \hat{N}_l,$$

$$s_k(x) = \sum_{n=1}^{N} \hat{r}_{nk} t(x_n), \qquad \hat{\lambda}_k = \lambda_0 + s_k(x). \tag{16}$$

We employ this coordinate ascent method to iteratively optimize the local and global parameters with the objective of maximizing the ELBO Hughes & Sudderth (2013).

## A.2 BIRTH AND MERGE MOVES HEURISTIC

In the moVB, we employ birth and merge moves as heuristic to attempt to escape from local optima and gain an global optimal target while updating the ELBO Hughes & Sudderth (2013).

Specially, the birth move involves 3 phases: **collection**, **creation**, and **adoption**.

1. In the **collection** phase, we gather a set of targeted sub samples, denoted as x', from the data focusing on a single component k'.
2. In the **creation** phase, we introduce new components by fitting a DP mixture model with K' components to x'. We expand the current model to include all K + K' components without immediately evaluating the change in ELBO, and always accept these additions. Subsequent merge moves are used to eliminate unnecessary components.
3. In the **adoption** phase, we reprocess all data batches and perform local and global parameter updates for the expanded K + K' component mixture. These updates involve expanded global summaries $S^o$, which incorporate summaries $S'$ from the targeted analysis of x'. This process leads to two interpretations of the subset x': assignment to original components (mostly k') and assignment to brand-new components. If the new components are favored, they gain mass through new assignments. After the pass, we subtract $S'$ to ensure both $S^o$ and global parameters align precisely with the data x. Any nearly-empty new component is likely to be pruned away by subsequent merges.

Similarly, the merge move comprises three steps: select components, form the candidate configuration q', and accept q' if the ELBO improves. After selecting the merged candidates $k_a$ and $k_b$ using a ratio of marginal likelihoods, our approach allows for the exact evaluation of the full-data ELBO to compare the existing configuration q with the merge candidate q' consisting of K-1 components. If the ELBO improves, we accept the proposal; otherwise, we reject it. The marginal likelihoods, donated as $M$, can be computed with cached summaries as follows:

$$p(k_b|k_a) \propto \frac{M(S_{k_a} + S_{k_b})}{M(S_{k_a})M(S_{k_b})}, \quad M(S_k) = exp\left(a_0(\lambda_0 + s_k(\mathbf{x}))\right). \tag{17}$$

## A.3 IMPLEMENTATION DETAILS

### A.3.1 VAE ARCHITECTURES

For image datasets, we use convolutional neural networks (Conv2d), as depicted in suppl. Figure 2. The output activation function of decoder is $tanh$, the reconstruction loss is MSE loss with mean reduction mode. Conversely, for text datasets, we adopt full-connection (FC) layers as Multi-layer-processing (MLP) module, where we directly use the linear output of decoder as reconstruction without activation node, the structure is shown in suppl. Figure 3.

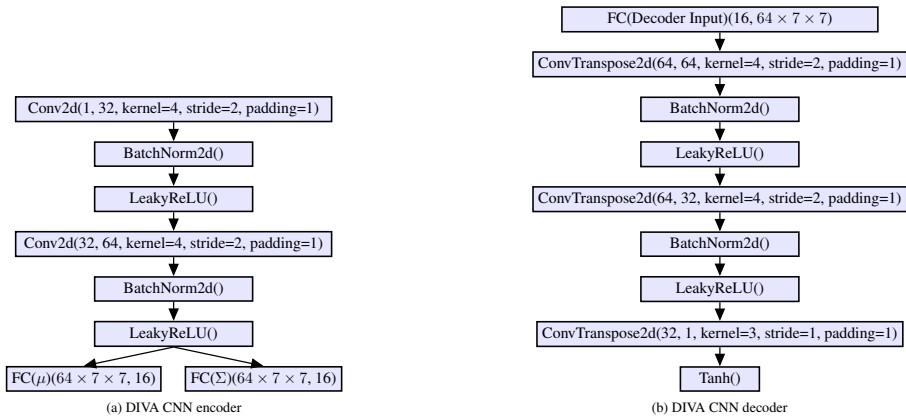

Figure 2: DIVA VAE Architecture for MNIST, Fashion-MNIST, CIFAR-10, SVHN

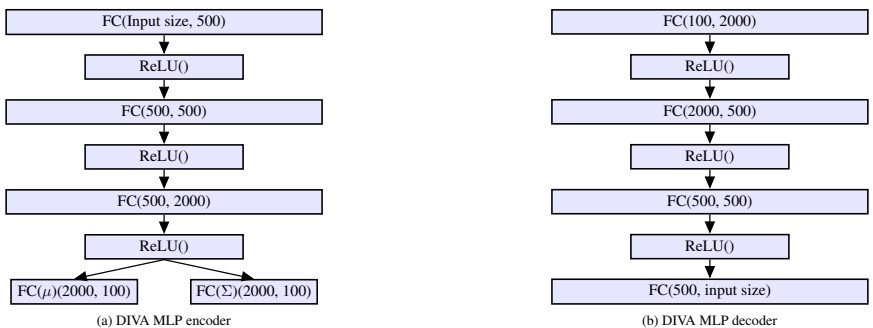

Figure 3: DIVA VAE Architecture for Reuters10k (input size 2000), HAR (input size 561), STL-10 (input size 2048), ImageNet (input size 128).

### A.3.2 HYPER-PARAMETERS

In addition, we summarize the hyper-parameters that commonly in use for all trials in the following suppl. Table 1.

Table 1: General key hyper-parameters of DIVA

| Description | Variable name | Value |
|---|---|---|
| learning rate | LR | $1e^{-3}, 2.5e^{-5}$ |
| learning rate decay | weight_decay | $0, 1e^{-4}, 1e^{-5}$ |
| train epoch(s) | max_epochs | max 500 |
| batch size | batch_size | 64, 128 |
| seed(s) | seed | 1, 2, 3, 4, 5 |
| KL divergence penalty factor | kld_weight | $1e^{-4}$ |
| reconstruction loss | [-] | MSE or NLL |
| optimizer | [-] | Adam |
| minimal number of atoms for new components | b_minNumAtomsForNewComp | 20, 80 |
| minimal number of atoms for target components | b_minNumAtomsForTargetComp | 40, 100 |
| minimal number of atoms for retain components | b_minNumAtomsForRetainComp | 40, 100 |
| scale factor of observe model | sF | 0.1, 0.01 |

### A.3.3 BASELINE FRAMEWORKS SETUP

**GMM**. We utilize the scikit-learn library Pedregosa et al. (2011) for GMM implementation. In all trials, we set the maximal iteration step to 100, while maintaining the covariance type of components as "*diag*", consistent with DIVA.

**GMVAE**. We directly utilize the officially released source code of GMVAE Dilokthanakul et al. (2016). The only modifications made are to the latent dimensions and the number of cluster components in the prior space. The training steps, batch size, and other hyperparameters remain consistent with DIVA. For further details, please refer to suppl. Table 1.

**DEC**. We leverage implementation of paper Xie et al. (2016).

**DPMM+memoVB**. We leverage implementation of paper Hughes & Sudderth (2013), the key hyperparameters are consistent with Table 1.

**VSB-DVM**. We leverage the implementation of paper Yang et al. (2019).

**DDPM**. Implementation is referred from paper Li et al. (2022a).

**DeepDPM**. We directly use the implementation of paper Ronen et al. (2022).

**SB-VAE**. We employ the VAE architecture of DIVA, incorporating a stick-breaking process to replace the Gaussian distribution in the latent space. The latent variable dimension and other hyperparameters remain consistent with DIVA, as detailed in suppl. Table 1.

## A.4 Full experimental results of DIVA

### A.4.1 Metrics

In this paper, we employ the widely used metric of unsupervised clustering accuracy (ACC) as one of our evaluation criteria. The metric has been extensively utilized in prior studies and is widely accepted in the field Jiang et al. (2017); Dilokthanakul et al. (2016). The formulation of the unsupervised clustering accuracy is presented below Jiang et al. (2017):

$$ACC = \max_f \frac{\sum_{i=1}^{N} \mathbf{1}\{l_i = f(v_i)\}}{N}, \tag{18}$$

where $f$ is a mapping from the learned DPMM clusters to the true class labels $l_i$. $N$ is the number of samples, $v_i$ is corresponding cluster assignment. Additionally, we also incorporate Normalized Mutual Information (NMI) and Adjusted Rand Index (ARI) as evaluation metrics. For all three metrics, the higher the better.

### A.4.2 Quantitative results of unsupervised clustering

The full test results on all datasets are shown in Table 2 and Table 3. mean $\pm$ (std.dev.) of 5 runs.

Table 2: Clustering performance on MNIST, Fashion-MNIST and Reuters10k. ‡: results from Xie et al. (2016).

| Frameworks | MNIST | | | Fashion-MNIST | | | Reuters10k | | |
|---|---|---|---|---|---|---|---|---|---|
| | ARI | NMI | ACC | ARI | NMI | ACC | ARI | NMI | ACC |
| GMM | $.19 \pm .02$ | $.30 \pm .02$ | $.60 \pm .01$ | $.35 \pm .05$ | $.51 \pm .01$ | $.49 \pm .02$ | $.35 \pm .05$ | $.41 \pm .03$ | $.73 \pm .06$ |
| DEC | $N/A$ | $N/A$ | $.84 \pm .00^{\ddagger}$ | $.45 \pm .01$ | $.53 \pm .02$ | $.60 \pm .04$ | $N/A$ | $N/A$ | $.72 \pm .00^{\ddagger}$ |
| GMVAE | $.57 \pm .04$ | $.79 \pm .03$ | $.82 \pm .04$ | $.44 \pm .02$ | $.57 \pm .01$ | $.61 \pm .01$ | $.40 \pm .12$ | $.42 \pm .09$ | $.73 \pm .08$ |
| DPMM+memoVB | $.13 \pm .01$ | $.39 \pm .01$ | $.63 \pm .02$ | $.23 \pm .03$ | $.48 \pm .01$ | $.57 \pm .01$ | $.17 \pm .03$ | $.26 \pm .05$ | $.56 \pm .05$ |
| VSB-DVM | $.66 \pm .07$ | $.75 \pm .04$ | $.86 \pm .01$ | $.41 \pm .01$ | $.57 \pm .01$ | $.64 \pm .03$ | $.31 \pm .07$ | $.34 \pm .08$ | $.60 \pm .03$ |
| DDPM | $.61 \pm .03$ | $.72 \pm .01$ | $.91 \pm .01$ | $.48 \pm .01$ | $.56 \pm .02$ | $.63 \pm .02$ | $.33 \pm .02$ | $.55 \pm .01$ | $.71 \pm .02$ |
| DeepDPM | $.91 \pm .02$ | $.90 \pm .01$ | $.93 \pm .02$ | $.52 \pm .01$ | $.68 \pm .01$ | $.63 \pm .01$ | $.62 \pm .01$ | $.67 \pm .01$ | $.83 \pm .01$ |
| **DIVA (Ours)** | $\mathbf{.73 \pm .04}$ | $\mathbf{.80 \pm .04}$ | $\mathbf{.94 \pm .01}$ | $\mathbf{.57 \pm .04}$ | $\mathbf{.83 \pm .01}$ | $\mathbf{.72 \pm .01}$ | $\mathbf{.68 \pm .07}$ | $\mathbf{.83 \pm .03}$ | $\mathbf{.83 \pm .01}$ |

Following tables display the unsupervised clustering accuracy on MNIST (suppl. Table 4), Fashion-MNIST (suppl. Table 5) and STL-10 (suppl. Table 6). Each data recorded as format "mean $\pm$ (std.dev.)". We run for each trial with 5 random seeds, and record its average value and corresponding standard deviation. To verify the model performance, we divide the datasets into train set and test set. For both MNIST and Fashion-MNIST datasets, we leverage torchvision implementation and use its default allocation ratio between train set and test set. For STL-10, we first extract data into low dimensional features using pretrained ResNet-50 and perform clustering on these low dimensional features.

Table 3: Clustering performance on HHAR, STL-10 and ImageNet-50

| Frameworks | HHAR | | | STL-10 | | | ImageNet-50 | | |
|---|---|---|---|---|---|---|---|---|---|
| | ARI | NMI | ACC | ARI | NMI | ACC | ARI | NMI | ACC |
| GMM | .26 ± .01 | .42 ± .01 | .43 ± .00 | .47 ± .01 | .60 ± .03 | .58 ± .03 | .32 ± .02 | .68 ± .00 | .60 ± .01 |
| DEC | .53 ± .01 | .65 ± .01 | .79 ± .01 | .46 ± .02 | .66 ± .03 | .80 ± .01 | .49 ± .01 | .70 ± .01 | .63 ± .01 |
| GMVAE | .28 ± .01 | .46 ± .02 | .65 ± .03 | .58 ± .04 | .76 ± .02 | .79 ± .04 | .47 ± .04 | .69 ± .01 | .62 ± .02 |
| DPMM+memoVB | .35 ± .02 | .57 ± .02 | .68 ± .04 | .51 ± .05 | .72 ± .02 | .64 ± .05 | .14 ± .00 | .65 ± .00 | .57 ± .00 |
| VSB-DVM | .52 ± .04 | .62 ± .06 | .66 ± .06 | .46 ± .01 | .62 ± .01 | .52 ± .03 | .39 ± .01 | .65 ± .01 | .49 ± .02 |
| DDPM | .63 ± .01 | .73 ± .03 | .74 ± .01 | .39 ± .02 | .57 ± .02 | .72 ± .01 | .50 ± .01 | .64 ± .01 | .63 ± .02 |
| DeepDPM | .65 ± .01 | .79 ± .01 | .79 ± .02 | .70 ± .01 | .79 ± .01 | .81 ± .02 | .55 ± .02 | .79 ± .02 | .66 ± .01 |
| **DIVA (Ours)** | **.71 ± .07** | **.86 ± .02** | **.83 ± .01** | **.87 ± .01** | **.95 ± .00** | **.72 ± .01** | **.88 ± .02** | **.97 ± .01** | **.69 ± .02** |

Table 4: Unsupervised clustering accuracy (%) on MNIST dataset

| # features | **DIVA (Ours)** | Vanilla DPMM | GMVAE K=10 | GMVAE K=5 | GMVAE K=3 | GMM K=10 | GMM K=5 | GMM K=3 |
|---|---|---|---|---|---|---|---|---|
| 3 features | .96 ± .01 | .85 ± .02 | .91 ± .01 | .96 ± .02 | .89 ± .05 | .77 ± .02 | .74 ± .02 | .74 ± .01 |
| 5 features | .95 ± .01 | .63 ± .01 | .95 ± .02 | .92 ± .02 | .60 ± .01 | .65 ± .03 | .48 ± .01 | .39 ± .01 |
| 7 features | .94 ± .01 | .60 ± .01 | .90 ± .01 | .68 ± .01 | .44 ± .01 | .64 ± .01 | .41 ± .00 | .31 ± .01 |
| 10 features | .94 ± .01 | .63 ± .02 | .82 ± .04 | .43 ± .01 | .30 ± .00 | .60 ± .01 | .33 ± .01 | .26 ± 0.00 |

Table 5: Unsupervised clustering accuracy (%) on Fashion-MNIST dataset

| # features | **DIVA (Ours)** | Vanilla DPMM | GMVAE K=10 | GMVAE K=5 | GMVAE K=3 | GMM K=10 | GMM K=5 | GMM K=3 |
|---|---|---|---|---|---|---|---|---|
| 3 features | .85 ± .01 | .78 ± .01 | .85 ± .01 | .90 ± .01 | .91 ± .01 | .85 ± .00 | .87 ± .01 | .66 ± .01 |
| 5 features | .71 ± .01 | .49 ± .00 | .70 ± .01 | .70 ± .01 | .55 ± .01 | .63 ± .01 | .57 ± .01 | .40 ± .01 |
| 7 features | .68 ± .01 | .40 ± .00 | .66 ± .01 | .56 ± .02 | .40 ± .01 | .59 ± .01 | .45 ± .00 | .38 ± .00 |
| 10 features | .72 ± .01 | .39 ± .01 | .61 ± .01 | .43 ± .01 | .29 ± .01 | .49 ± .02 | .35 ± .00 | .28 ± .01 |

Table 6: Unsupervised clustering accuracy (%) on STL-10 dataset

| # features | **DIVA (Ours)** | Vanilla DPMM | GMVAE K=10 | GMVAE K=5 | GMVAE K=3 | GMM K=10 | GMM K=5 | GMM K=3 |
|---|---|---|---|---|---|---|---|---|
| 3 features | .99 ± .00 | .86 ± .00 | .99 ± .01 | .99 ± .01 | .99 ± .01 | .91 ± .00 | .98 ± .01 | .88 ± .01 |
| 5 features | .96 ± .01 | .82 ± .00 | .94 ± .01 | .84 ± .03 | .60 ± .01 | .89 ± .01 | .88 ± .01 | .58 ± .01 |
| 7 features | .93 ± .01 | .81 ± .01 | .86 ± .01 | .70 ± .00 | .43 ± .01 | .80 ± .01 | .60 ± .04 | .41 ± .01 |
| 10 features | .88 ± .01 | .70 ± .01 | .79 ± .04 | .49 ± .01 | .30 ± .01 | .58 ± .03 | .46 ± .01 | .28 ± .01 |

### A.4.3 DYNAMIC ADAPTATION

Suppl. Figure 4 shows the DIVA t-SNE snapshots of dynamic adaptation test on MNIST with incremental feature training. Notably, DIVA can birth new components when the new inputs are introduced in the subset.

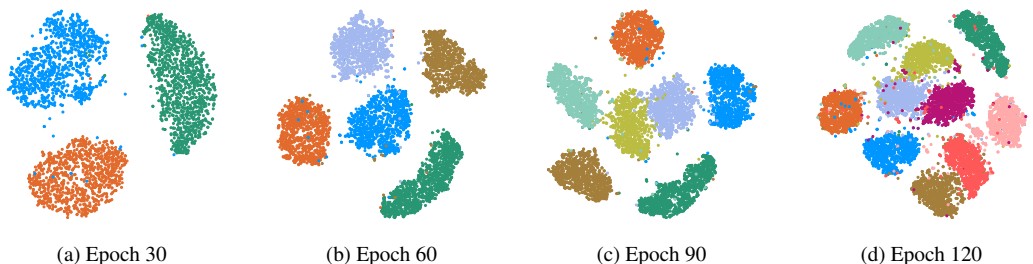

(a) Epoch 30      (b) Epoch 60      (c) Epoch 90      (d) Epoch 120

Figure 4: t-SNE snapshots of DIVA on MNIST dynamic adaptation test at epoch (a) 30, (b) 60, (c) 90 and (d) 120. The training curve refer to main page Fig. 6.

Suppl. Figure 5 shows the "birth & merge" moves of DIVA during training on MNIST dataset. We conduct 4 trials, where in the first 3 trials the model is trained under static case, which means the number of feature is unchanged during training and stable at 3, 5, 10 respectively. In the 4-the trial, the model is trained with dynamically increased features, where the feature number increases at epoch 30, 60, 90 from initial 3 to 5, 7, 10 respectively. It is worth noting that under static case, the DIVA births the appropriate number of clusters to fit the observation, once the training converges, the

redundant clusters may merge to one to improve the ELBO of the inference. In dynamic changing feature case, the similar results could be acquired, the number of clusters may increase according to the increased feature number.

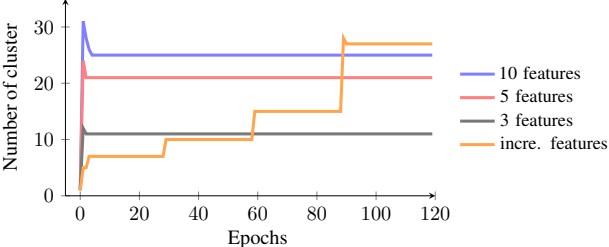

Figure 5: "Birth & Merge" moves of DIVA on MNIST. We conduct 4 trials, in which the number of features in the first 3 trials are static at 3, 5, 10 respectively. In the 4-th trial, the number of feature is initially at 3 and increases at epoch 30, 60, 90 to 5, 7, 10 respectively.

### A.4.4    RECONSTRUCTION IMAGES OF DIVA LEARNED CLUSTERING COMPONENTS

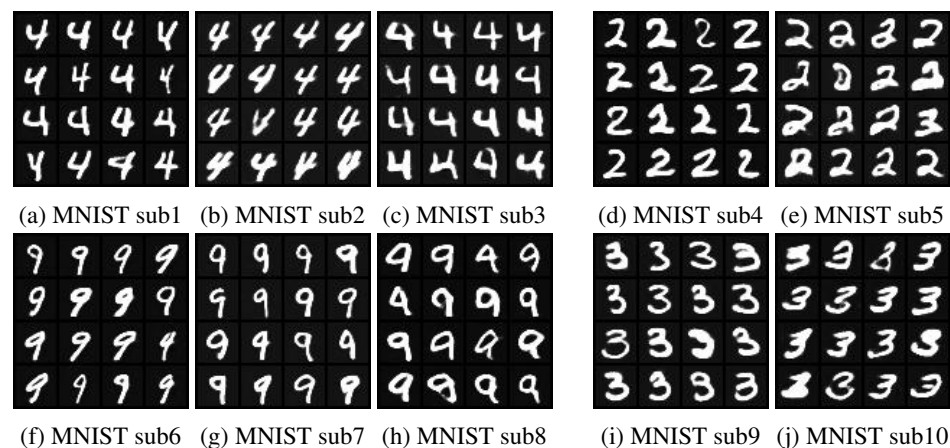

Figure 6: Reconstruction images from DPMM learned clusters on MNIST. We employ a 2-layer CNN as backbone of nonlinear structure (suppl. Fig. 2). It is noting that our proposed framework DIVA can efficiently extract informative sub-features from coarse labels.

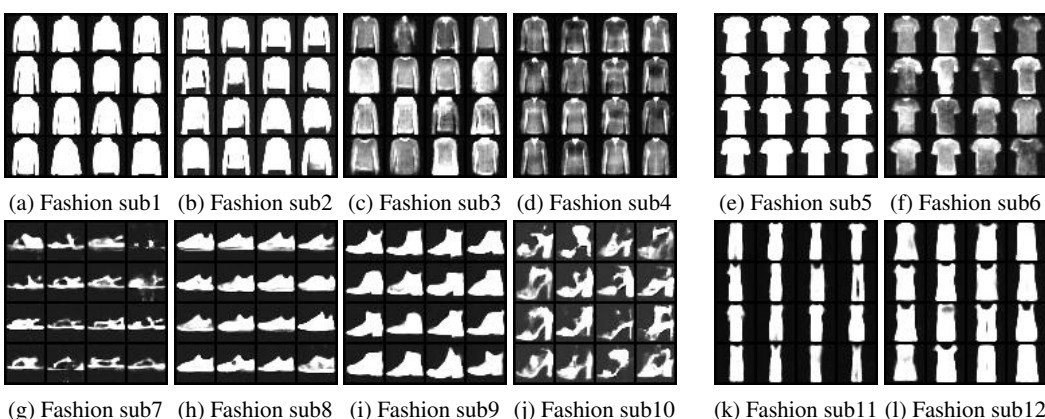

Figure 7: Reconstruction images from DPMM learned clusters on Fashion-MNIST. We employ a 2-layer CNN as backbone of nonlinear structure (suppl. Fig. 2). It is noting that our proposed framework DIVA can efficiently extract informative sub-features from coarse labels.

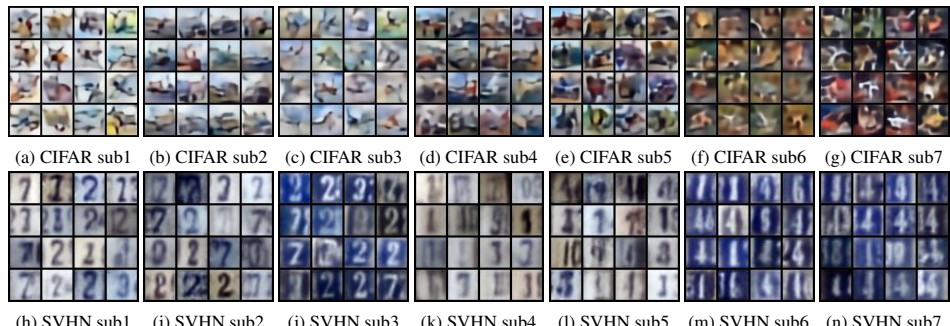

(a) CIFAR sub1  (b) CIFAR sub2  (c) CIFAR sub3  (d) CIFAR sub4  (e) CIFAR sub5  (f) CIFAR sub6  (g) CIFAR sub7

(h) SVHN sub1  (i) SVHN sub2  (j) SVHN sub3  (k) SVHN sub4  (l) SVHN sub5  (m) SVHN sub6  (n) SVHN sub7

Figure 8: Reconstruction images from DPMM learned clusters on CIFAR-10 (a)-(g) and SVHN (h)-(n). We employ a 2-layer CNN as backbone of nonlinear structure (suppl. Fig. 2). It is noting that our proposed framework DIVA can efficiently extract informative sub-features from coarse labels.

## A.5 TRACEABILITY OF DPMM CLUSTERING LEARNING

One remarkable characteristic of DIVA clustering is its traceability, wherein a cluster's index position in the list and mapping to its corresponding ground truth label remain unchanged as long as the cluster does not undergo "birth" or "merge" moves. In our study, each individual cluster component in the DPMM is represented by a multivariate diagonal Gaussian distribution with parameter pairs $\{\mu_k, \Sigma_k\}_{k=0:K}$. The management of DPMM clustering is facilitated by manipulating the number, position and value of these parameters within a list object.

To provide a clear illustration, we present a simplified case in Suppl. Figure 9. Initially, DIVA is trained and converged on the MNIST dataset with only two ground truth labels, "0" and "1" (Suppl. Figure 9a), resulting in three clusters in the DPMM. Cluster 0 captures label "1", while clusters 1 and 2 capture label "0".

In the subsequent stage, we extend the dataset range to include three ground truth labels by adding the digits "2" to the dataloader (Supplementary Figure 9b). The original clusters in the DPMM fail to adequately represent the new features in the dataset. Consequently, during the fitting process, the DPMM generates five new clusters to capture the new features: three clusters learn digit "2", and two clusters learn digit "0". Subsequently, the DPMM attempts to merge redundant clusters to enhance the performance based on the ELBO. By merging the clusters 1 and 2 from the initial stage, which learned digit "0", the original cluster 2 disappears, and the new merged cluster occupies index 1. As a result, all other newly formed clusters shift one position forward to positions 2...6, as shown in Suppl. Figure 9b.

However, the original cluster 0 from the initial stage does not participate in any birth or merge operations, thus maintaining its position in the management list and its mapping to ground truth label "1" unchanged.

## A.6 SHUFFLE THE CLUSTERS MAY IMPROVE PERFORMANCE

In addition to the birth and merge proposals, we also introduce the "shuffle" moves as an alternative in the fitting process of DPMM. After the birth and merge moves in each fitting lap, the "shuffle" move can be optionally activated, which rearranges the clusters based on the number of data points belonging to each active component and re-update the global parameters. This reordering places clusters with a larger number of data points at the forefront. This strategy can be beneficial for improving the ELBO by updating the global parameters and fast identifying potential more suitable candidates for birth and merge moves within a limited number of laps. Consequently, it may lead to an overall improvement in the clustering performance of DIVA. Supplementary Figure 10 illustrates an example of DIVA's test accuracy on MNIST with and without the shuffle functionality. It is observed that incorporating shuffle moves leads to slightly improved performance compared to the setup without shuffle moves. Ultimately, the accuracy is improved by approximately 3% at the end of training.

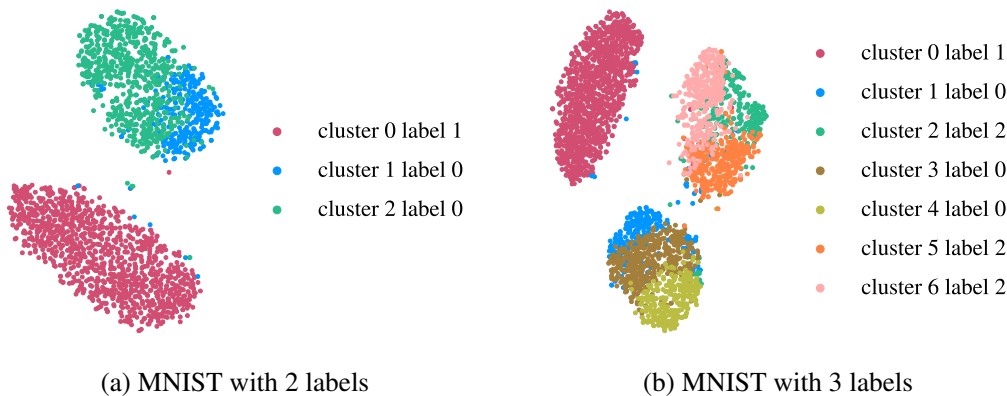

(a) MNIST with 2 labels         (b) MNIST with 3 labels

Figure 9: The t-SNE projection of MNIST with (a) 2 ground truth labels, (b) 3 ground truth labels, colored by clusters. Moving from subfigure (a) to (b), DIVA gains access to an additional label, namely "2". During the fitting process, DIVA first generates 5 new clusters to accommodate the new observations. In the merging phase, the original clusters 1 and 2 in subfigure (a) merge into the new cluster 1 in subfigure (b). Simultaneously, all other newly formed clusters shift by one position, becoming clusters 2 to 6 in subfigure (b). Notably, cluster 0 remains unaffected by the birth and merge operations, maintaining its mapping to its corresponding captured ground truth and position intact within the cluster list.

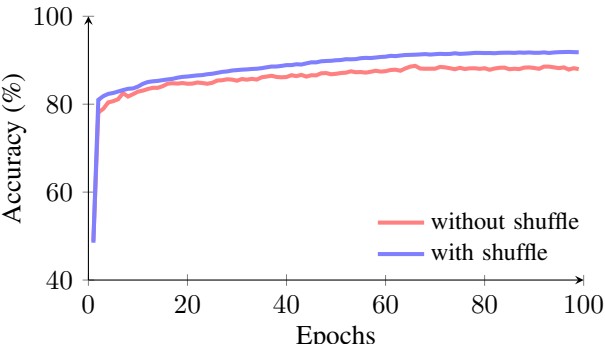

Figure 10: The test accuracy of DIVA trained on MNIST full dataset with additional "shuffle" moves (blue), and without "shuffle" moves (red).

