# OpenReview forum: "DIVA: A Dirichlet Process Mixtures Based Incremental Deep Clustering Algorithm via Variational Auto-Encoder"
_ICLR.cc/2024/Conference — ICLR 2024 Conference Withdrawn Submission_

### Official Review · Reviewer_Ggf7 · 2023-10-14

**Soundness:** 2 fair
**Presentation:** 2 fair
**Contribution:** 2 fair
**Rating:** 3
**Confidence:** 5

**Summary:**

The paper presents a VAE where the encoder [posterior over the latent representation] is not a Gaussian but a Dichlet process mixture of Gaussians.

**Strengths:**

The paper combines some well-known techniques, namely Gaussian VAE and the memoized VB algorithm for Dirichlet process mixture models. There are some experiments with small datasets, which at least are known benchmarks of the old time.

**Weaknesses:**

The concept of the method is inconsistent and, to my mind, signals a superficial understanding of the field. We used to employ nonparametrics because a single μ and Σ estimate cannot capture the variability of the data. However, when your μ and Σ are parametric functions of the data [here, deep networks parameterizing the VAE / encoders] there is absolutely NO need for nonparametrics. You are not dealing with a single μ  and Σ estimate, but as many as the data points. Besides, note that typically in nonparametrics we used to set K (the number of atoms) equal to the number of data points, which was considered "close to infinity" and, of course, has no place in a deep net-parameterized VAE. Therefore, the method lacks any substance actually.

Novelty-wise, it is just a superficial blend of existing algorithms.

**Questions:**

how does the method behave close to "infinity", and not with a superficially low number of K?

---

### Official Review · Reviewer_Hi2d · 2023-10-28

**Soundness:** 2 fair
**Presentation:** 2 fair
**Contribution:** 3 good
**Rating:** 3
**Confidence:** 4

**Summary:**

This paper proposes DIVA, a novel method for clustering high-dimensional datasets using variational autoencoders. DIVA is a Dirichlet Process Mixture model fitted on the latent space of a VAE, and under variational inference assumptions can be optimised through its ELBO. Its key, claimed innovation is to alleviate the user from having to choose the number of clusters a priori, which DIVA learns adaptively from data.

**Strengths:**

* This method provides a, to the best of my knowledge, novel method.
* The method solves an indeed open and important problem in the deep clustering literature, how to adaptively choose and learn the number of clusters from data, without requiring this to be set through prior knowledge of a user.
* The background provided is informative, and the mathematical formulation of the method is solid.
* The combined method in Algorithm 1 is non-trivial, combines several techniques for optimising the model, and is of interest to the community.

**Weaknesses:**

* My main concern is that the method’s promise of alleviating the need to know the number of parameters a priori is only to some degree satisfied. While the number of Gaussian components are learned from the data, this number is determined by other hyperparameters of the model. In particular, I would like to ask the authors to comment on all hyperparameters that the method introduces and that need to be set, and specifically comment on the selection and effect of the concentration parameter alpha.
* The authors evaluate their method experimentally mainly using the clustering accuracy metric (ACC) to cluster as the supervised label, and qualitatively with t-SNE. The former evaluation is limited: Image datasets typically contain more than one valid partition [2], and focussing on the one imposed by the supervised label is in the best case limited, in the worst case inappropriate. – This paper requires further evaluation, for instance in terms of its generative capability and the diversity of generated samples (generating actual samples, see also my point below), but also provide more details into what clusters are actually learned. On the latter point, Figures 7 and Figures 6 and 7 in the Appendix only show a subset of the learned clusters. It remains unclear if they are cherry-picked, what all the clusters learn, how the shown examples are sorted, and how many clusters there are in total.
* The writing is sometimes hard to understand or unclear and could be improved. There are frequent grammatical issues.
* What is stated as “Generative performance” are not samples from the VAE, but merely reconstructions. This should be changed, or samples should additionally be provided, which would give additional insight into the model. – To see what is learned in each cluster, visualising the clustered input examples would be sufficient.
* A minor point: Very established VAE work has the acronym DIVA which is unfortunate, namely “Ilse, M., Tomczak, J.M., Louizos, C. and Welling, M., 2020, September. Diva: Domain invariant variational autoencoders. In Medical Imaging with Deep Learning (pp. 322-348). PMLR.” I suggest picking a different name for the method, or no name at all.

I am happy to reconsider my current recommendation upon receiving a response to the above points.

**Questions:**

* Table 1 should include the results of VaDE [1] and MFCVAE [2], where available. For instance on MNIST, VaDE (94.46 %) “outperforms” DIVA (94.01%) and is likewise within range of standard deviation of MFCVAE, questioning the claim by the authors that their method “outperforms state-of-the-art baselines”. – In particular, I would like to ask the authors why they decided to not report VaDE even though they were aware of and mentioning it in the related work.
* DIVA does not support multi-facet clustering (in comparison to MFCVAE), which should be stated as a limitation.
* Eq. (1), left-most looks odd. Could you please explain? It seems inconsistent with the graphical model.
* I would be interested in the active number of clusters on the experimental datasets.

[1] Jiang, Z., Zheng, Y., Tan, H., Tang, B. and Zhou, H., 2016. Variational deep embedding: An unsupervised and generative approach to clustering. arXiv preprint arXiv:1611.05148.

[2] Falck, F., Zhang, H., Willetts, M., Nicholson, G., Yau, C. and Holmes, C.C., 2021. Multi-facet clustering variational autoencoders. Advances in Neural Information Processing Systems, 34, pp.8676-8690.

---

### Official Review · Reviewer_pC7Y · 2023-10-31

**Soundness:** 2 fair
**Presentation:** 2 fair
**Contribution:** 2 fair
**Rating:** 5
**Confidence:** 3

**Summary:**

The paper proposes DIVA, a Dirichlet Process Mixtures Incremental deep clustering framework via Variational Auto-Encoder (VAE). DIVA assumes soft clustering of observations in the latent space via a learnable Dirichlet Process Mixture Model (DPMM) prior, where the VAE and DPMM prior parameters are learned iteratively. DIVA is a non-parametric clustering approach, where the number of clusters is unknown. Experimental results on six datasets (including ImageNet-50) demonstrate that DIVA outperforms baselines (parametric/non-parametric) in dynamic and static datasets.

**Strengths:**

- The paper is relatively well-written and easy to follow
- DIVA outperforms baselines in static and dynamic datasets, crucial for lifelong learning applications

**Weaknesses:**

- DIVA seems like a straightforward combination of DPMM and VAE
- It's unclear how DIVA handles the "death" and "merge" process of clusters. I encourage the authors to include complete details in the main paper
-  It seems DIVA is comparable to DeepDPM in the static settings. The DeepDPM setup for dynamic datasets doesn't seem fair.  I encourage the authors to include an ablation study, e.g., replace DPMM with DeepDPM objective function

**Questions:**

- What is the computational efficiency of DIVA relative to alternative non-parametric baselines, e.g., DeepDPM?
- Could you provide results on inferred K relative to ground truth, for all non-parametric methods?
- For static datasets, it seems DeepDPM is comparable to DIVA. Could you clarify why DeepDPM fails in the dynamic setup?
- Figure 6: Could you clarify why GMVAE (K=10) seems to degrade in performance over time?
- Figure 4: Shouldn't we expect GMVAE (K=10) to have a similar latent structure as DIVA?
- Figure 6: Could you provide dynamics on all datasets besides MNSIT?